# Miocene Tropical Forests in South China Shaped by Combined Asian Monsoons

**DOI:** 10.3390/plants14233599

**Published:** 2025-11-25

**Authors:** Hao Zhang, Robert A. Spicer, Cheng Quan, Luliang Huang, Jianhua Jin

**Affiliations:** 1State Key Laboratory of Biocontrol and Guangdong Provincial Key Laboratory of Plant Stress Biology, School of Life Sciences, Sun Yat-sen University, Guangzhou 510275, China; alwayszhh1108@126.com; 2Yunnan Key Laboratory of Forest Ecosystem Stability and Global Change, Xishuangbanna Tropical Botanical Garden, Chinese Academy of Sciences, Mengla 666303, China; r.a.spicer@open.ac.uk; 3State Key Laboratory of Tibetan Plateau Earth System, Resources and Environment (TPESRE), Institute of Tibetan Plateau Research, Chinese Academy of Sciences, Beijing 100101, China; 4School of Environment, Earth and Ecosystem Sciences, The Open University, Milton Keynes MK7 6AA, UK; 5School of Earth Science and Resources, Chang’an University, Xi’an 710054, China; quan@chd.edu.cn; 6Nanjing Institute of Geology and Palaeontology, Chinese Academy of Sciences, Nanjing 210008, China

**Keywords:** Asian monsoon, paleobotany, paleoclimate, paleoecology, South China

## Abstract

The Miocene epoch witnessed the emergence of modern biomes and biodiversity hotspots. Understanding its history in South China is crucial for informing conservation under modern climate change, yet quantitative constraints on the evolution of climate and vegetation from the tropical–subtropical transition zone remain scarce. Here, we present the first quantitative Miocene paleoclimatic and paleoecological reconstructions based on an integrated analysis of leaf-based proxies applied to exceptionally preserved and highly diverse dicotyledonous leaf megafossils from the Erzitang Formation, Guiping Basin, Guangxi. Results indicate a mean annual temperature of 22.3 °C and mean annual precipitation of 1991 mm, with a monsoon intensity index higher than present, indicating a humid monsoonal climate regime. Vegetation analysis identifies the Miocene Guiping flora as tropical forest. Rather than a simple forest replacement, South China maintained dynamic tropical forest patches that expanded northward to 23° N under Asian monsoons, forming a mosaic with evergreen broad-leaved forests. Overall, the Miocene Guiping vegetation represents a tropical forest situated in a tropical rainforest to seasonal forest ecotone under a humid monsoonal climate, rather than a per-humid rainforest, underscoring the pivotal role of monsoon evolution in shaping low-latitude forest patterns and providing a deep-time benchmark for predicting vegetation responses to future climate change.

## 1. Introduction

Influenced by the East Asian monsoon, the tropical and subtropical regions in China host one of the great global biodiversity hotspots, characterized by a rich variety of terrestrial biomes and ecosystem types. Thus, understanding of the origin and evolution of biodiversity in this region is of great significance for guiding the protection of biodiversity under the impact of current global climate change. The Miocene epoch has been heralded as marking the origins of modern terrestrial biomes, as well as many of the world’s biodiversity hotspots [1,2,3,4,5]. In China, phylogenetic reconstructions of the angiosperm flora indicate that 66% of extant genera originated no earlier than the Early Miocene (~23 Ma) [6], but this figure requires revision in the light of new dating of many key fossil taxa from southwestern China [7]. A multidisciplinary consensus is emerging that the evolution of the East Asian monsoon was the decisive driver of floristic diversification and the rise in evergreen broad-leaved forests (EBLFs) across the region [8].

Support for this scenario comes from integrative phylogenomic analyses of the *Litsea* complex (Lauraceae)—a key lineage that includes dominant species of EBLFs—which combine molecular data with geological, paleontological, and Cenozoic climate records [9]. These studies show that the development of a strong monsoon over East Asia in the Early Miocene [10,11] accelerated the evolution of evergreen habits, ultimately precipitating the modernization of EBLFs in East Asia [9]. Paleobotanical evidence, however, reveals pronounced spatial heterogeneity in this process [8,12]. Although evergreen broad-leaved forests likely occurred in southwestern China during the Miocene [13], coeval deposits from southern Fujian in southeastern China were occupied by tropical rainforest vegetation comparable to that of present-day Southeast Asia [14,15,16]. The Zhangpu biota of Fujian captures a mid-Miocene megathermal seasonal rainforest that extended to 24.2° N and rivaled modern Southeast Asia in diversity [16]. The biota contains a rich and exquisitely preserved fossil arthropod fauna and abundant inclusions of plants, fungi, snails, and even feathers, all in amber, forming a species-rich community whose genera are still extant—an “evolutionary museum” [16]. The leaf-physiognomy analysis of fossil leaves associated with the amber biota yields a mean annual temperature of 22.5 °C, with a warmer winter and precipitation seasonality comparable with present-day conditions [16]. In canonical correspondence analysis (CCA) space, the flora plots within the East Asian monsoon realm but abuts the South Asian monsoon region, implying that monsoons and winter warming during the Mid-Miocene Climatic Optimum (MMCO) drove the northward expansion of tropical rainforest into southeastern China [16].

South China, situated between these contrasting domains, occupies a crucial yet enigmatic transitional zone. It presents as a typical tropical-to-subtropical monsoon region of China, characterized by high temperatures and abundant precipitation: the annual number of days with a daily mean temperature ≥10 °C exceeds 300, and mean annual precipitation ranges from 1400 to 2000 mm [17]. Precipitation exhibits pronounced inter-annual variability, marked intra-annual uneven distribution, and obvious regional heterogeneity [17]. The vegetation spectrum is highly diverse, encompassing evergreen coniferous forest, evergreen broad-leaved forest, deciduous broad-leaved forest, mixed forest, and grassland [17]. Despite its pivotal position within the climatic and ecological transition zone, systematic characterization of the Miocene climate and vegetation dynamics in this region has hitherto been largely lacking.

The Guiping Basin in Guangxi preserves an exceptionally thick and fossiliferous Miocene succession that has already yielded diverse tropical–subtropical plant taxa—including members of Cibotiaceae [18], Podocarpaceae [19,20], Burseraceae [21], Ceratophyllaceae [22], Elaeocarpaceae [23], Juglandaceae [24], and Altingiaceae [25]—yet quantitative paleoclimatic parameters and refined vegetation–environment relationships for this basin are still lacking.

Studies of leaf-climate have always been of great interest [26,27,28,29,30,31]. Leaves are sessile organs directly exposed to the ambient environment; through natural selection their morphology is tuned to maximum photosynthetic benefits with minimal economic loss [32,33,34]. In woody dicotyledons, this adaptive variation mirrors the prevailing climate [32,35,36]. Paleoclimate proxies based on leaf fossil morphology utilize this adaptability to transform the leaf morphological characteristics preserved in fossils into quantitative reconstructions of past climates [36]. 

Despite the growing body of evidence for monsoon-influenced vegetation dynamics in East Asia, quantitative paleoclimatic and paleoecological data from the tropical–subtropical transition zone in South China remain scarce. In this study, we aim to fill this gap by presenting the first quantitative reconstruction of the Miocene climate and vegetation for the Guiping Basin, based on an integrated analysis of exceptionally preserved dicotyledonous leaf megafossils. Our objectives are to (1) estimate key paleoclimatic parameters, including temperature and precipitation, (2) characterize the paleovegetation type and its ecological structure, and (3) assess the role of monsoon evolution in shaping low-latitude forest patterns in South China during the Miocene.

## 2. Results

### 2.1. Paleoclimate Estimation

Low taxonomic diversity can compromise the application of the leaf margin analysis (LMA). Although there is no consensus on the minimum number of taxa required for a reliable LMA, uncertainty generally declines as the number of taxa increases [37]. Uhl et al. (2003) similarly showed that the accuracy of LMA temperature estimates largely depends on the number of taxa and the quantity of fossil specimens [38]. Their analyses considered regions with at least 15 taxa. Following this standard, we selected 46 fossil taxa from the Guiping Basin (Figure 1, Figure 2 and Figure 3), approximately 87% of which exhibited entire acted margins. Applying the Chinese regression equation to the leaf fossils of the Guiping Basin yielded a calculated mean annual temperature (MAT) of 26.1 ± 1.25 °C (Table 1).

The precipitation estimate from LAA is based on the average leaf area of 42 fossil taxa (Appendix A). Substituting the average leaf area into the global regression equation resulted in a MAP estimate of 1108.1 mm (670.92–3916.74 mm) (Table 1).

We scored 31 leaf character states for 46 fossil taxa in the Guiping Basin and employed the standard CLAMP procedure (Appendix A). The CLAMP analysis provided a completeness statistic of 0.71, which is above the reliability threshold of 0.66 [31], indicating that our results are likely accurate. The most dominant characteristics were that 87% of the taxa had entire margins, followed by 76.5% of the taxa possessing acute leaf bases, 56.8% of the taxa having elliptical leaf shapes, and 50.6% of the taxa with a length-to-width ratio of 2–3:1. None of the taxa exhibited features such as lobed leaves, compound teeth, leaf sizes belonging to nanophyll, leptophyll I, and leptophyll II, or length-to-width ratios less than 1:1. Figure 4 illustrates the structure of the CLAMP morphospace. The Guiping flora is situated within the East Asian monsoonal vegetation realm yet, in all three dimensions, lies immediately adjacent to the South Asian monsoonal vegetation and its transitional area (Figure 4). To estimate paleoclimate, regression models were constructed for each climate vector in the axis 1–3 space, and the position of the fossil site was plotted on the regression.

The first variable to consider is MAT, which was estimated at 22.3 ± 2.27 °C. The WMMT variable was 27 ± 2.3 °C, while the CMMT was 17 ± 3.35 °C, indicating seasonal temperature differences. The LGS was 11.7 ± 1.04, suggesting that temperatures remained above 10 °C throughout the year. Considering GSP is defined as the time temperatures are ≥10 °C, this implies year-round growth potential. In terms of precipitation, the Miocene precipitation in the Guiping Basin was relatively high, with GSP estimated at 1991 ± 481 mm.

The hydroclimate also exhibited seasonal variations, with 3-WET at 974 ± 237.2 mm, approximately 5.4 times that of 3-DRY (180 ± 57.5 mm). This difference between the 3-WET and 3-DRY variables indicates the presence of seasonal precipitation differences. Despite this, humidity variables show less seasonal contrast. The PET_ann_div10 for the Guiping Basin during the Miocene was 137.3 ± 14.95 mm, while PET_wrm was 134.9 ± 18.55 mm, both somewhat higher than PET_cld (83.7 ± 13.13 mm). The high paleoenvironmental PET—especially in summer—implied stronger natural evapotranspiration forcing, such as elevated temperatures and potentially higher wind strength, reflecting higher temperatures and drier summer conditions. However, in terms of absolute amounts of water in the atmosphere, VPD is remarkably uniform. VPD_ann for the Guiping Basin during the Miocene was 6.4 ± 2.02 hPa, with VPD_spr at 8.5 ± 2.74 hPa, VPD_sum at 5 ± 3.09 hPa, VPD_aut at 5.7 ± 1.79 hPa, and VPD_win at 5.8 ± 1.30 hPa, indicating atmospheric moisture was highest in the summer, but throughout the year, there was never a very dry atmosphere (Table 1).

### 2.2. Integrated Plant Record (IPR) Vegetation Analysis

The Miocene Erzitang Formation of the Guiping Basin yields a rich plant-fossil assemblage that typifies tropical–subtropical flora (Appendix A). Across all classification protocols, broad-leaved evergreen (BLE) taxa dominate 60% (Appendix A). Under the Mix-Drudge-1 protocol, BLE accounted for 76.92%, broad-leaved deciduous (BLD) only 15.38%, and sclerophyll + legume (SCL + LEG) and herbaceous components (D-HERB, M-HERB) were absent. The Mix-Drudge-2 shows a similar pattern: BLE taxa decline modestly to 62.50%, BLD taxa remain low at 12.50%, and all other categories were recorded as zero. Collectively, the Guiping assemblage was characterized by “high evergreen, low deciduous, and no sclerophyllous or herbaceous elements”. Integrated analyses of both IPR and taxonomic similarity indicate that the Miocene Guiping flora is most closely allied to modern tropical rainforests, with a total difference of 45.38% (Appendix A).

## 3. Discussion

### 3.1. Comparison of Paleoclimate Reconstruction Methods

A series of Miocene MAT estimates for the Guiping Basin is summarized in Table 1. Related to the multivariate method, the univariate LMA yields notably higher MAT (26.1 ± 1.25 °C) than CLAMP (22.3 ± 2.27 °C). One explanation is that the LMA calibration excludes other leaf traits that carry temperature signals [39], which often leads to MAT overestimation at low-latitude sites [40]. Given the complex coupling between leaf traits and climate, and the profound influence of biogeographic history, LMA has increasingly been superseded by multivariate methods as a climate proxy [39]; its parameter instability across regions and through time undermines the reliability [41].

Miocene MAP estimates for the Guiping Basin were also inferred using different methods (Table 1). CLAMP does not return MAP estimates but instead estimates GSP, i.e., precipitation during months with temperatures ≥10 °C. Because the inferred growing season for Guiping is 11.7 months, the GSP estimate is effectively equivalent to MAP. In contrast, LAA gives a MAP of 1108.14 mm (670.92–3916.74 mm), lower than the CLAMP estimate of 1991 ± 481 mm. Although leaf size covaries with MAP, LAA carries substantial uncertainties. Peppe et al. (2011) found that LAA tends to underestimate MAP at high-precipitation sites and overestimate it at low-precipitation sites, indicating that LAA is a useful tool for roughly estimating MAP, but not a precise estimator [42].

### 3.2. A Humid Monsoonal Climate with Warm Winters and Evergreen Dominance During the Miocene in Guiping

During the Miocene, Guiping recorded a MAT of 22.3 °C—comparable to the present—while CMMT reached 17 °C, ~4 °C warmer than today, indicating frost was rare in winter. MAP approached 2000 mm, with more than 970 mm falling within the three wettest months and still ca. 180 mm during the three driest months. MSI was ~40. Specifically, the CLAMP CMMT of ~17 °C lies below the Köppen Af threshold (≥18 °C); the MSI of ~40 indicates moderate precipitation seasonality. CLAMP morphospace analysis (Figure 4) placed the Guiping flora within the Asian monsoonal vegetation realm, but immediately adjacent—in all three dimensions—to transitional vegetation, underscoring its pivotal position at the Miocene interface of the East Asian and South Asian monsoon systems. Together these data indicate a humid monsoonal climate characterized with warm winters, abundant precipitation, and moderate seasonality. This regime contrasts sharply with coeval sites at the same latitude in Yunnan, southwestern China, where both CMMT and MSI were markedly lower [40,43,44,45]. In contrast, the mid-Miocene flora of Zhangpu, Fujian, southeastern China, exhibits temperature and precipitation ranges that almost overlap those of Guiping [16]. Taken together, it is implied that the lowlands of southern China lay under the influence of the evolving monsoon systems and formed the tropical margin of the Asian monsoon.

Additionally, IPR analysis showed that the Miocene Guiping flora was distinct within South China because of its exceptionally high broad-leaved evergreen forest (BLE) component of 76.92%. (Appendix A).

### 3.3. Persistence and Northward Expansion of Miocene Tropical Elements in Guiping

South China was one of the first regions in East Asia to host EBLFs. Paleobotanical evidence shows that drought-tolerant taxa dominated from the Paleocene to the Early Eocene, whereas a major environmental shift by the Middle Eocene produced subtropical, humid evergreen–deciduous broad-leaved mixed forests [46,47]. The Middle Eocene assemblages from Changchang and Maoming in South China, as well as the late Eocene floras from Japan, also exhibited EBLFs characteristics dominated by evergreen Fagaceae and Lauraceae [35,48,49,50,51]. Genomic analyses place the prototype of East Asian EBLFs in the early Eocene (55–50 Ma), fostered by ”greenhouse” warming [9], whereas a crown-based lineage accumulation rate (LAR) study of 72 diagnostic genera dates the origin of extant Asian EBLFs to no later than the early Oligocene (~31.8 Ma) [52]. Model and proxy data converge to indicate that the Eocene Asian experienced a unique bimodal precipitation regime with wet winters, promoting the modernization of East Asian biodiversity [53]. High-precision chronological and paleoclimatic analysis of 41 Ma strata from central Yunnan by Fang et al. (2021) have been used to suggest a pulsed northward jump of the Indian monsoon in the late Eocene, driving the tropical monsoon front from <20° N to 26° N [54]; however, definitive evidence that this was genuinely the Indian monsoon, or even a monsoon regime at all, is lacking.

Phylogenomic study further indicates that cooling and aridification across subtropical East Asia between ~48 to ~38 Ma promoted the evolution of deciduous habits within dominant EBLFs lineages; subsequently, the intensification of the East Asian monsoon system in the Early Miocene (~23 Ma) increased precipitation, accelerated the re-emergence of evergreen habits of the dominant lineages, and ultimately shaped the vegetation into forms resembling those of today [9]. A key insight from the Guiping Basin refines this narrative: the Miocene flora at Guiping is classified as ”tropical rainforest” (Appendix A) whereas application of Whittaker’s biome scheme to the site’s climate parameters places it in a tropical seasonal forest (Appendix A) [55,56,57,58]. Specifically, the Guiping Miocene point falls on the boundary between tropical seasonal forest/savanna and tropical rainforest, lying closer to the tropical seasonal forest zone than to the core tropical rainforest zone. This discrepancy likely reflects the contrast between climate-envelope classifications (which emphasize temperature and precipitation interactions) and floristic diagnostics. While Whittaker’s model captures climatic controls, it can yield ambiguous assignments in transitional zones, and it gives limited weight to compositional and structural attributes, leading to occasional mismatches between climatic classes and realized vegetation. Moreover, high BLE alone is not sufficient to discriminate tropical rainforest from tropical monsoon forest or subtropical EBLFs. Overall, the Miocene Guiping vegetation is best regarded as tropical forest, occupying a tropical rainforest to seasonal forest ecotone under a warm winter, abundant precipitation, and moderate seasonal monsoon regime, rather than a per-humid Af rainforest.

This site provides a strong case that complicates the conventional narrative of a simple, unidirectional replacement of tropical forests by evergreen broad-leaved forests. Although the fundamental architecture of EBLFs was established by the Early Miocene, tropical forests not only persisted in South China but—even more strikingly—expanded northward to 23° N. Fossiliferous Middle Miocene strata in Fujian, southeastern China, corroborate this trajectory: they contain typical tropical elements such as Dipterocarpaceae [14,15,16], further documenting a rainforest-to-EBLF dynamic [8]. Southwestern China may have followed a distinct Miocene pathway [8]. Collectively, these findings show that the coastal and interior lowlands of southeastern China and South China were warm, humid, and frost-free, representing the northernmost limit of tropical rain and monsoon forests; the southwestern region, uplifted by topography, experienced cooler winters and supported evergreen broad-leaved forests [13,59].

Quantitative paleoclimate reconstructions from Guiping and Zhangpu jointly reveal that Miocene South China experienced a “high rainfall plus warm winter” regime driven by the coupled development of the East Asian and South Asian monsoons—a climatic envelope now restricted to the lowlands of the Indochina Peninsula. Guiping, therefore, provides a critical, high-resolution case study that (i) constrains the latitudinal gradient of monsoon seasonality and (ii) illustrates how tropical forests and EBLFs maintained a long-term, non-linear balance in response to monsoon evolution.

## 4. Conclusions

This study presents the first quantitative paleoclimatic and paleoecological reconstruction of the Miocene Erzitang Formation from the Guiping Basin, South China, based on an integrated analysis of exceptionally preserved dicotyledonous leaf megafossils. Our results indicate a humid monsoonal climate with a mean annual temperature of 22.3 °C and mean annual precipitation of approximately 1991 mm. The vegetation is best characterized as tropical forest, occupying a dynamic ecotone between tropical rainforest and seasonal forest, rather than a per-humid rainforest environment.

The Guiping flora exhibits a high proportion of broad-leaved evergreen taxa and lacks sclerophyllous or herbaceous elements, reinforcing its tropical affinity. The reconstructed climate parameters, including a warm winter (CMMT ~17 °C), and moderate precipitation seasonality alongside the CLAMP morphospace suggest that South China during the Miocene was influenced by the combined effects of the East Asian and South Asian monsoon systems. This monsoonal regime facilitated the northward expansion of tropical forest elements to at least 23° N, forming a mosaic with evergreen broad-leaved forests across the region.

These findings suggest a more complex scenario than the traditional view of a unidirectional replacement of tropical forests by evergreen broad-leaved forests in East Asia. Instead, they support a more nuanced model in which tropical forests persisted and even expanded in lowland South China during the Miocene, coexisting with subtropical vegetation types. This study highlights the critical role of monsoon evolution in shaping low-latitude forest patterns and provides a deep-time benchmark for understanding vegetation responses to future climate change.

## 5. Materials and Methods

### 5.1. Geological Setting

The Erzitang Formation of the Guiping Basin, and the fossil site (23°23′09.67″ N, 110°09′55.21″ E; elevation, 49 m) is in Xunwang Town near Guiping City, Guangxi Zhuang Autonomous Region (Figure 5). The fossil host sediments mainly consist of lacustrine and swamp deposits composed of yellow and red mudstones (Figure 5) [60]. Based on mammalian fossils (*Prolipotes yujiangensis* Zhou, Zhou, & Zhao) and plant fossils (*Quercus* sp.), the age is generally considered to be Miocene [60]. Previous studies have followed this age assignment [18,19,20,21,22,23,24,25,61], but absolute date constraints are lacking.

### 5.2. Research Methods

Here, we present paleoclimatic and paleoecological reconstructions for the Guiping Basin based on a comprehensive census of exceptionally preserved dicotyledonous leaf megafossils and taxa represented by other organs. We document more than twenty angiosperm leaf morphotypes (MT), defined by leaf apex shape, leaf margin type, leaf base shape, and leaf venation features, providing a robust dataset for applying a range of leaf physiognomic proxies. By converting these traits into quantitative estimates of key climatic and ecological variables and assigning taxa to ecologically informative taxonomic–physiognomic components, we both fill a critical gap in the regional record of EBLF evolution and elucidate the role of South China as a dynamic ecotone responding to the intensification of the Asian monsoon system during the Miocene.

#### 5.2.1. LMA

Leaf margin analysis (LMA) is based on the positive correlation between the percentage of entire-leaved species and the mean annual temperature (MAT) [27]. This relationship can be used to infer past MAT. The specific steps are as follows: (1) Collect data on the percentage of entire-leaved species in modern vegetation and MAT to establish a correlation. (2) Use this correlation to convert the percentage of entire-leaved species in fossil plant communities into an estimated MAT. However, this correlation varies across regions [62]. Su et al. (2010) systematically collected data from 50 sample sites in China to establish a Chinese regression equation, improving local accuracy [62]. Furthermore, Chen et al. (2014) argued that their Chinese regression equation, with a larger sample size, provides more accurate paleotemperature results [63]. Given the geographical location of the fossil site, we adopted the equation based on the Chinese database proposed by Chen et al. (2014) [63] (Equation (1)) to calculate the estimated MAT.(1)MAT=0.223E+6.68,
where *MAT* is the estimated mean annual temperature derived from leaf margin analysis; and *E* is defined as the proportion of entire-margined woody dicotyledonous plant species at a site relative to the total number of sampled species.

Additionally, Miller et al. (2006) incorporated the uncertainty associated with the proportion of toothed-leaved species at a site into the uncertainty assessment of MAT estimates (Equation (2)) [64].(2)$$\sigma [MAT]=c\surd \{[1+\varphi (n-1)E(1-E)]\frac{E(1-E)}{n}\},$$
where *c* is the slope of the regression between MAT and leaf margin in the dataset used; *E* is the proportion of entire-margined species; *n* is the number of sampled species; and *φ* is the over-dispersion factor for *E*, calculated by Miller et al. (2006) as 0.052 [64].

#### 5.2.2. LAA

Leaf area analysis (LAA) is a method that estimates the mean annual precipitation (MAP) using leaf size (leaf area). It is based on Bailey and Sinnott’s (1915, 1916) early observation that large-leaved plants are more common in warm and humid environments [65,66]. Subsequent studies found a positive correlation between leaf size and MAP. Givnish (1984) first quantitatively studied the correlation between leaf size and MAP in plant communities of South America, Central America, and Australia [34]. Wilf et al. (1998) expanded on Givnish’s work by developing predictive equations using the average natural logarithm of leaf area (MlnA) and the natural logarithm of MAP (lnMAP) from 50 sites across North America, Central America, South America, and Africa [67]. The relationship between leaf size and MAP was validated in multiple plant communities, including those in Bolivia, temperate regions of the Northern Hemisphere, equatorial Africa, and plant communities in North America, Central America, South America, Asia, and Oceania, leading to the development of a series of predictive equations [42,68,69,70,71].

In this study, we used the predictive equations developed by Peppe et al. (2018) based on global plant communities (Equations (3) and (4)) to calculate MAP [71].(3)ln(MAP)=0.346MlnA+2.404,(4)$$MlnA=\sum a_{i}p_{i},$$
where *a_i_* represents the average natural logarithm of leaf area for each size category, typically based on true leaf size; and *p_i_* is the proportion of morphotypes in each category.

#### 5.2.3. CLAMP

CLAMP is a freely accessible program (http://clamp.ibcas.ac.cn/) (accessed on 24 March 2025) that utilizes at least 20 woody dicotyledonous angiosperm species and 31 leaf character states, such as leaf shape, size, margin type, or tooth morphology, to estimate paleoclimate variables using canonical correspondence analysis as its statistical foundation [28,30,31,39,72,73,74]. Currently, CLAMP can provide estimates for 23 paleoclimate variables. Unlike more univariate proxies referred to above, the CLAMP physiognomic data are derived from the field following strict protocols, and not from herbarium data that may not reflect the full morphological variation present in leaf form at a given location under a known climate.

In this study, we first scored the 31 leaf character states for 46 leaf morphotypes according to the method described on the CLAMP website. The estimates for 23 paleoclimate variables were then determined based on the PhysgAsia2AZ and WorldClim2_GRIDMetAsia2AZ_22var datasets. The PhysgAsia2AZ dataset was verified in geological use by oxygen isotopes and is suitable for paleoaltimetry studies in Asia [75,76,77,78]. It includes modern vegetation data exposed to the Asian monsoon [76,79]. The WorldClim2_GRIDMetAsia2AZ_22var is a high-resolution climate calibration file based on WorldClim2 climate data (1970–2000) with data extracted for the same locations as the PhysgAsia2 dataset [80].

The fossil site of the Guiping Basin was positioned passively within CLAMP morphospace defined by these calibration datasets to estimate paleoclimate variables related to leaf morphological characteristics. These variables include mean annual temperature (MAT), cold month mean temperature (CMMT), warm month mean temperature (WMMT), length of growing season (LGS), growing season precipitation (GSP), precipitation of the three wettest months (3-WET), precipitation of the three driest months (3-DRY), specific humidity (SH.ann), moist enthalpy (ENTH), mean minimum temperature during the warmest month (MinT_W), mean maximum temperature during the coldest month (MaxT_C), mean annual vapor pressure deficit (VPD_ann), mean vapor pressure deficit during the three summer months (VPD_sum), mean vapor pressure deficit during the three winter months (VPD_win), mean vapor pressure deficit during the spring months (VPD_spr), mean vapor pressure deficit during the three autumn months (VPD_aut), mean annual potential evapotranspiration (PET_ann), mean potential evapotranspiration during the warmest month (PET_wrm), and mean potential evapotranspiration during the coldest month (PET_cld).

Leaves adapted to monsoons exhibit a spectrum of character traits associated with different monsoon types [35,36,39]. Canonical correspondence analysis (CCA) can position fossil assemblages within the trait space to reveal which monsoon (or non-monsoon) climate type the fossil trait spectrum is most adapted to CCA [35]. By comparing the leaf trait spectrum of the fossil assemblage with those of modern leaves grown under monsoon and non-monsoon conditions, the climate experienced by the fossil leaves can be attributed to a particular monsoon (or non-monsoon) type [35]. The monsoon seasonality intensity (MSI) can be measured using this index, proposed by Xing et al. (2012), which can be calculated based on climate variables returned by CLAMP (Equation (5)) [45].(5)MSI=(3WET−3DRY)×100/GSP,

The higher the value of this index, the greater the difference in precipitation between the wet and dry seasons, indicating a stronger monsoon seasonality [81]. 

#### 5.2.4. Integrated Plant Record Vegetation Analysis

We employed Integrated Plant Record (IPR) vegetation analysis [82] to perform a semi-quantitative assessment of the fossil plant assemblages, and thereby reconstruct their corresponding zonal vegetation type. The IPR method integrates megafossils (leaves, fruits, seeds, wood) and palynological records, assigns taxa to ecologically informative taxonomic–physiognomic components, and classifies the resultant spectra into zonal vegetation categories [83,84].

All plant taxa (presence–absence only) were allocated to the following components: Zonal woody components: CONIFER, broad-leaved deciduous (BLD), broad-leaved evergreen (BLE), sclerophyllous/legume-like (SCL + LEG), zonal palms (ZONPALM), and arborescent ferns (ARBFERN). Zonal herbaceous components: xerophytic herbs (DRY HERB) and mesophytic herbs (MESO HERB). Non-zonal components: azonal woody (AZONAL WOODY), azonal non-woody (AZONAL NON-WOODY), aquatic plants (AQUATIC), and problematic taxa (PROBLEMATIC TAXA) [84].

We calculated (i) the ratio of zonal woody angiosperm components (BLD + BLE + SCL + LEG) to total zonal woody angiosperms (BLD + BLE + SCL + LEG + ZONPALM), and (ii) the ratio of zonal herbs (DRY HERB + MESO HERB) to all zonal taxa [85]. Based on these proportions, each assemblage was assigned to one of eight zonal vegetation types: temperate to warm-temperate broad-leaved deciduous forest (BLDF); warm-temperate to subtropical mixed mesophytic forest (MMF); subtropical evergreen broad-leaved forest (BLEF); BLDF–MMF ecotone; MMF–BLEF ecotone; subtropical sub-humid sclerophyllous/microphyllous forest (ShSF); open woodland (OW); and xeric grassland (Xeric GRASS) [84]. To ensure reliability, only assemblages containing ≥15 zonal taxa were analyzed [86].

Fossil IPR parameters were compared to a reference dataset of 505 modern vegetation plots [85] using the Drudge 1/2 tool to compute IPR similarity (Euclidean distance) and generic taxonomic similarity, thereby identifying the closest modern analogs [87]. All analyses were conducted via the IPR vegetation database online platform (http://www.iprdatabase.eu) (accessed on 15 August 2025) [85].

## Figures and Tables

**Figure 1 plants-14-03599-f001:**
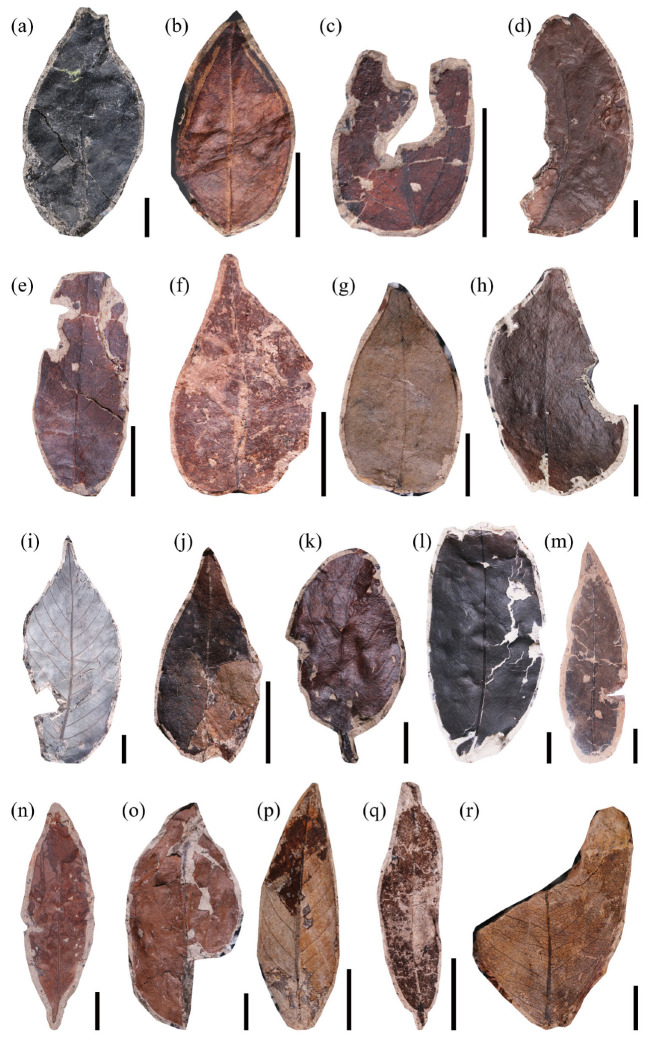
Leaf morphotypes of the Guiping Basin (1), scale bar = 10 mm: (**a**) morphotype 1, specimen number GP-495; (**b**) morphotype 2, specimen number GP-498; (**c**) morphotype 3, specimen number GP-505; (**d**) morphotype 4, specimen number GP-506; (**e**) morphotype 5, specimen number GP-515; (**f**) morphotype 6, specimen number GP-524; (**g**) morphotype 7, specimen number GP-531; (**h**) morphotype 8, specimen number GP-537; (**i**) morphotype 9, specimen number GP-928; (**j**) morphotype 10, specimen number GP-540; (**k**) morphotype 11, specimen number GP-546; (**l**) morphotype 12, specimen number GP-556; (**m**) morphotype 13, specimen number GP-569; (**n**) morphotype 14, specimen number GP-572; (**o**) morphotype 15, specimen number GP-599; (**p**) morphotype 16, specimen number GP-600; (**q**) morphotype 17, specimen number GP-601; (**r**) morphotype 18, specimen number GP-605.

**Figure 2 plants-14-03599-f002:**
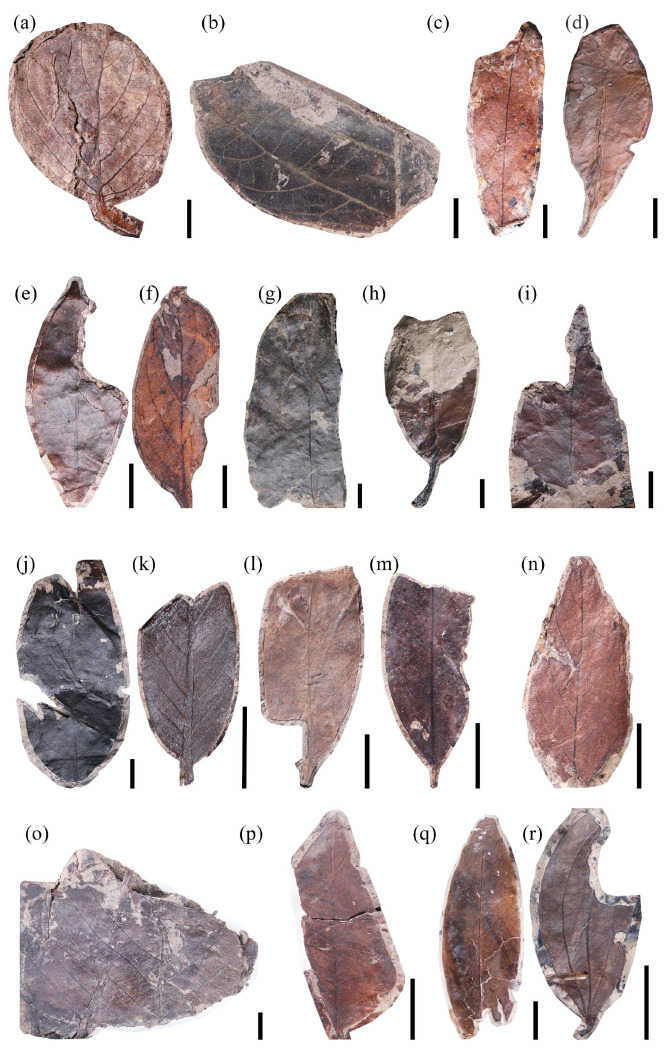
Leaf morphotypes of the Guiping Basin (2), scale bar = 10 mm: (**a**) morphotype 19, specimen number GP-849; (**b**) morphotype 20, specimen number GP-770; (**c**) morphotype 21, specimen number GP-772; (**d**) morphotype 22, specimen number GP-773; (**e**) morphotype 23, specimen number GP-784; (**f**) morphotype 24, specimen number GP-796; (**g**) morphotype 25, specimen number GP-799; (**h**) morphotype 26, specimen number GP-802-A; (**i**) morphotype 26, specimen number GP-802-B; (**j**) morphotype 27, specimen number GP-815; (**k**) morphotype 28, specimen number GP-871; (**l**) morphotype 29, specimen number GP-882; (**m**) morphotype 30, specimen number GP-906; (**n**) morphotype 31, specimen number GP-919; (**o**) morphotype 32, specimen number GP-921; (**p**) morphotype 33, specimen number GP-942; (**q**) morphotype 34, specimen number GP-946; (**r**) morphotype 35, specimen number GP-959.

**Figure 3 plants-14-03599-f003:**
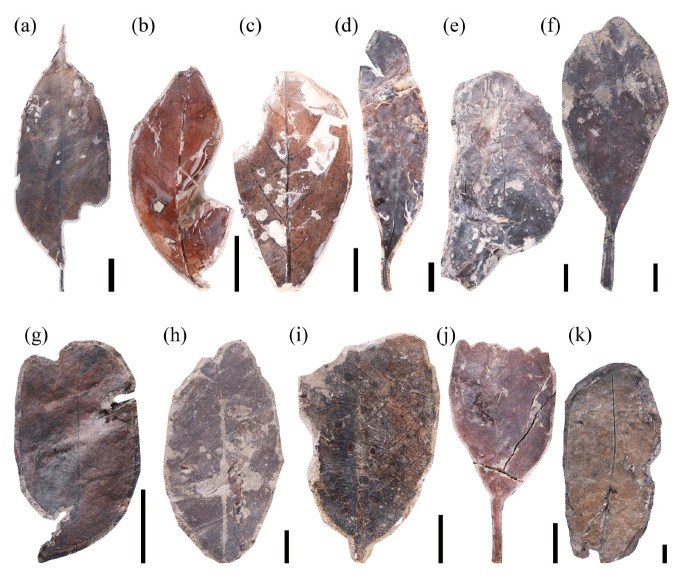
Leaf morphotypes of the Guiping Basin (3), scale bar = 10 mm: (**a**) morphotype 36, specimen number GP-954; (**b**) morphotype 37, specimen number GP-967; (**c**) morphotype 38, specimen number GP-969; (**d**) morphotype 39, specimen number GP-971; (**e**) morphotype 40, specimen number GP-972; (**f**) morphotype 46, specimen number GP-1008; (**g**) morphotype 41, specimen number GP-988; (**h**) morphotype 42, specimen number GP-995; (**i**) morphotype 43, specimen number GP-996; (**j**) morphotype 44, specimen number GP-997; (**k**) morphotype 45, specimen number GP-1001.

**Figure 4 plants-14-03599-f004:**
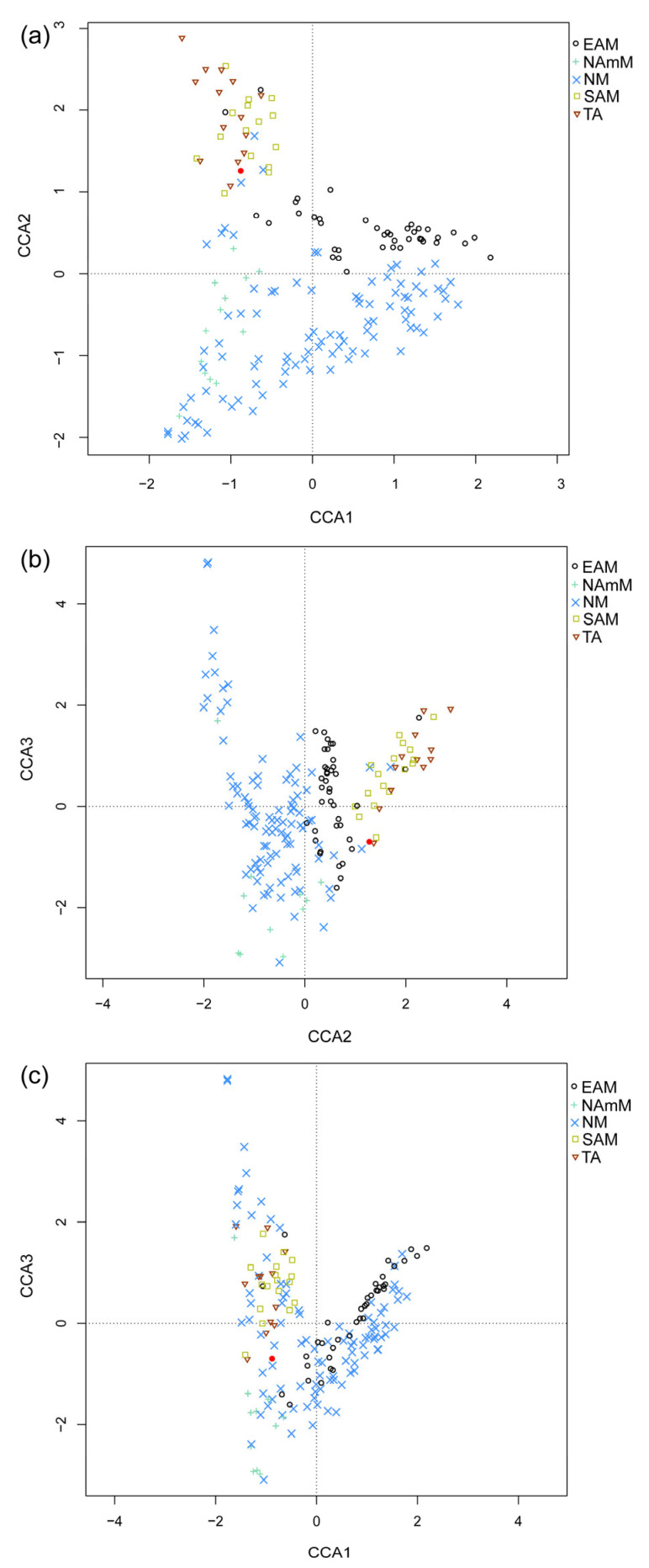
CLAMP plots showing the position of leaf fossil flora from the Guiping Basin within the PhysgAsia2 calibration space. Red dots indicate the fossil flora, while modern vegetation sites code for the monsoon type to which they are exposed (EAM—East Asia Monsoon, NAmM—North America Monsoon, NM—No Monsoon, SAM—South Asia Monsoon, TA—Transitional Area). The figure uses three canonical correspondence analysis (CCA) plots—axes 1 vs. 2 (**a**), axes 2 vs. 3 (**b**), and axes 1 vs. 3 (**c**)—to visualize relationships between data points. The fossil flora falls within the PhysgAsia2 physiognomic space, confirming the dataset’s suitability for reconstructing the paleoclimate of Guiping. In the CLAMP morphospace, the fossil flora (Axis 1 = −0.894, Axis 2 = 1.258, Axis 3 = −0.700) plots closest to the TA cluster (Euclidean distance = 1.59), followed by SAM (1.91) and EAM (1.93), confirming its pivotal position at the East Asian and South Asian monsoon interface.

**Figure 5 plants-14-03599-f005:**
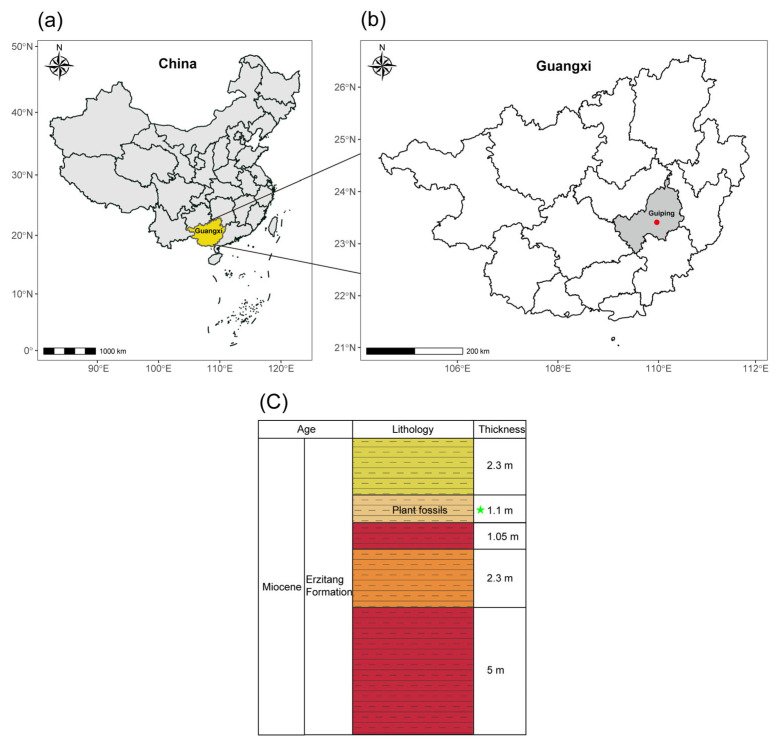
The geological background of the leaf fossil locality. (**a**,**b**) A schematic diagram showing the geographical location of the leaf fossil locality marked with a red solid circle; (**c**) a simplified stratigraphic column of the Erzitang Formation in the Guiping Basin showing age, lithology, and thickness. The succession is dominated by red–brown, red, and yellow mudstones with cumulative thicknesses ranging from 1.05 m to 5 m. The plant fossil horizon is marked with a green solid star.

**Table 1 plants-14-03599-t001:** The modern and Miocene climate of Guiping Basin estimated using leaf margin analysis (LMA), leaf area analysis (LAA), and Climate-Leaf Analysis Multivariate Program (CLAMP).

Variables	Units	Guiping (Modern)	Guiping (Fossil)		
			LMA	LAA	CLAMP
MAT	°C	22.1	26.1 ± 1.25		22.3 ± 2.27
MAP	mm	1687.4		1108.1	
WMMT	°C	29.3			27 ± 2.3
CMMT	°C	13			17 ± 3.35
LGS	Months	12			11.7 ± 1.04
GSP	mm	1687.4			1991 ± 481
3-WET	mm	742.7			974 ± 237.2
3-DRY	mm	154.3			180 ± 57.5
SH_ann	g/kg	13.67			13.5 ± 1.63
ENTH	kJ/kg	34.81			34.8 ± 0.79
VPD_ann	hPa	5.96			6.4 ± 2.02
VPD_sum	hPa	7.63			5 ± 3.09
VPD_win	hPa	3.9			5.8 ± 1.3
VPD_spr	hPa	4.78			8.5 ± 2.74
VPD_aut	hPa	7.53			5.7 ± 1.79
PET_ann_div10	mm				137.3 ± 14.95
PET_wrm	mm				134.9 ± 18.55
PET_cld	mm				83.7 ± 13.13
MinT_W	°C				22.9 ± 2.47
MaxT_C	°C				22.3 ± 3.39
MSI	%	34.87			39.88

Modern values are from https://www.paleo.bristol.ac.uk/cgi-bin/date_umpart01.cgi (accessed on 15 March 2025). Climatic parameters are as follows: mean annual temperature (MAT), mean annual precipitation (MAP), warm month mean temperature (WMMT), cold month mean temperature (CMMT), length of growing season (LGS), growing season precipitation (GSP), three wettest months’ precipitation (3-WET), three driest months’ precipitation (3-DRY), specific humidity (SH), moist enthalpy (ENTH), mean annual vapor pressure deficit (VPD_ann), mean vapor pressure deficit during the three summer months (VPD_sum), mean vapor pressure deficit during the three winter months (VPD_win), mean vapor pressure deficit during the spring months (VPD_spr), mean vapor pressure deficit during the three autumn months (VPD_aut), mean annual potential evapotranspiration (PET_ann), mean potential evapotranspiration during the warmest month (PET_wrm), mean potential evapotranspiration during the coldest month (PET_cld), mean minimum temperature during the warmest month (MinT_W), and the mean maximum temperature during the coldest month (MaxT_C). The growing season is defined as the period measured in months during which the mean temperature is ≥10 °C. With the CLAMP growing season being effectively 12 months, the GSP (1991 ± 481 mm) is equivalent to the MAP.

## Data Availability

The original contributions presented in this study are included in the Appendix A. Further inquiries can be directed to the corresponding authors.

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
