# Peer review of "Miocene Tropical Forests in South China Shaped by Combined Asian Monsoons"

_plants, 2025, doi:10.3390/plants14233599_

Round 1

Reviewer 1 Report

Comments and Suggestions for Authors

The paper reconstructs Miocene paleoclimate and vegetation for the Guiping Basin using multiple leaf-based proxies (Leaf Margin Analysis, Leaf Area Analysis, CLAMP) and IPR-based vegetation pattern. It argues for a high-rainfall, warm-winter monsoonal regime and a long-term tropical forest–EBLF coexistence. The results are potentially meaningful. I suggest it will be published after major revisions.

The LMA-based inference (estimated from petiole width²/leaf area) to infer the leaf economic spectrum and, by extension, vegetation type, the analysis is not decision-useful here: the usable sample is very small with many singletons for per morphotype, and confidence intervals are wide. I recommend removing this section.

CLAMP morphospace (Figure 4): I recommend indicating in the figure caption the fossil point’s coordinates and/or its nearest-neighbor distances to the EAM, SAM, and transition clusters to facilitate reader interpretation.

In lines 335-337: the authors hold the view “Miocene vegetation in Guiping can be classified as ‘tropical rainforest’ (Table S4).” The evidence presented is insufficient to support a definitive assignment to tropical rainforest. The floristic/physiognomic signal (e.g., high BLE proportions in IPR) suggests rainforest affinity, but key climatic diagnostics and robustness checks are missing or ambiguous. Specifically, the CLAMP-derived CMMT (~17 °C) falls below the Köppen tropical threshold (coldest-month ≥ 18 °C), while MSI (~40) indicates moderate seasonality and the CLAMP morphospace places Guiping at the East–South Asian monsoon interface. These features are consistent with a rainforest–monsoon-forest ecotone and do not uniquely identify Af sensu Köppen. High BLE alone is not sufficient to discriminate tropical rainforest from tropical monsoon forest or subtropical EBLFs.

Lines 337–339. The manuscript states: “By contrast, according to Whittaker’s biome classification model [58–60], the climate parameters of Guiping during the Miocene indicate the presence of tropical monsoon forests.” Please substantiate this claim with a Whittaker biome placement figure: plot the Guiping Miocene point(s) in MAT–MAP space, overlay standard biome boundaries, and show uncertainty (e.g., 95% CI/error bars). Please provide the underlying values and the boundary source in the SI.

The manuscript states that “This single site overturns the conventional narrative…”; this is rhetorically strong for site-level evidence. Please soften to “provides a strong complicates the narrative”.

Comments on the Quality of English Language

Several sections are overly verbose, which obscures the central points. An annotated PDF with suggestions has been provided. Please undertake a careful revision.

Reviewer 2 Report

Comments and Suggestions for Authors

Dear Authors,
Your manuscript entitled „Miocene Tropical Forests in South China Shaped by Combined Asian Monsoons” contains interesting findings. In my opinion it might interest an international audience therefore it is worth publishing. Generaly the manuscript is well-planned and written. Nevertheless, I have found some imperfections, which (in my opinion) should be imperved or at least clarified before an eventual publication. I have listed the m below:
1.    I suggest to add main conclusion in Abstract section.
2.    Text from lines 97-106 better suits to chapter Material and methods. In my opinion chapter Introduction should be ended by specification of study aims.
3.    Tables should be selfexplanatory. The meaning of all abbreviations and acronymes must be explained in captions.
4.    Figures 4 and 5 in present form is hardly legible. Their quality must be improved.
5.    Presented results are really valuable. The chapter Conclusions should be added. Plese, poin out the novelty of Your findings and indicate directions of future studies. 

Round 2

Reviewer 1 Report

Comments and Suggestions for Authors

No